TARGE: large language model-powered explainable hate speech detection

Hashir Muhammad Haseeb 1
Memoona 1
http://orcid.org/0000-0001-8454-6980 Kim Sung Won 2 swon@yu.ac.kr
1 Information and Communication Engineering, Yeungnam University , Gyeongsan, Gyeongbuk , Republic of South Korea
2 School of Computer Science and Engineering, Yeungnam University , Gyeongsan, Gyeongbuk , Republic of South Korea
Zubiaga Arkaitz
Electronic publication date: 2025 May 30
Publication date: 2025
Volume: 11
Electronic Location ID: e2911
Received 2024 Dec 17; Accepted 2025 Apr 30
Copyright: © 2025 Hashir et al.
Copyright year: 2025
Copyright holder: Hashir et al.
License: This is an open access article distributed under the terms of the Creative Commons Attribution License, which permits unrestricted use, distribution, reproduction and adaptation in any medium and for any purpose provided that it is properly attributed. For attribution, the original author(s), title, publication source (PeerJ Computer Science) and either DOI or URL of the article must be cited.
License URL: https://creativecommons.org/licenses/by/4.0/

Keywords: Social media, Hate speech, Large language models, Rationale extraction

Funding: Basic Science Research Program through the National Research Foundation of Korea (NRF) Ministry of Education NRF-2021R1A6A1A03039493 Korean government (MSIT) NRF-2022R1A2C1004401 This study was funded by the Basic Science Research Program through the National Research Foundation of Korea (NRF), the Ministry of Education (NRF-2021R1A6A1A03039493) and by the NRF grant funded by the Korean government (MSIT) (NRF-2022R1A2C1004401). There was no additional external funding received for this study. The funders had no role in study design, data collection and analysis, decision to publish, or preparation of the manuscript.

==============================
The proliferation of user-generated content on social networking sites has intensified the challenge of accurately and efficiently detecting inflammatory and discriminatory speech at scale. Traditional manual moderation methods are impractical due to the sheer volume and complexity of online discourse, necessitating automated solutions. However, existing deep learning models for hate speech detection typically function as black-box systems, providing binary classifications without interpretable insights into their decision-making processes. This opacity significantly limits their practical utility, particularly in nuanced content moderation tasks. To address this challenge, our research explores leveraging the advanced reasoning and knowledge integration capabilities of state-of-the-art language models, specifically Mistral-7B, to develop transparent hate speech detection systems. We introduce a novel framework wherein large language models (LLMs) generate explicit rationales by identifying and analyzing critical textual features indicative of hate speech. These rationales are subsequently integrated into specialized classifiers designed to perform explainable content moderation. We rigorously evaluate our methodology on multiple benchmark English-language social media datasets. Results demonstrate that incorporating LLM-generated explanations significantly enhances both the interpretability and accuracy of hate speech detection. This approach not only identifies problematic content effectively but also clearly articulates the analytical rationale behind each decision, fulfilling the critical demand for transparency in automated content moderation.

Introduction

Social media networks have revolutionized global interaction, establishing virtual forums where participants across societal, ethnic, and regional divides converge to share perspectives and knowledge. However, these digital spaces, while fostering unprecedented connectivity, can deteriorate into venues for antagonistic discourse and discriminatory rhetoric. The concept of hate speech encompasses intentional public expressions designed to marginalize or degrade specific demographics based on inherent characteristics. Including, but not exclusively, ethnic identity or sexual orientation (Nockleby, 1994; Perera et al., 2023). The ramifications of online hate speech extend beyond virtual boundaries, manifesting in tangible societal harm. A stark illustration of this phenomenon emerged amid the COVID-19 outbreak, Findling et al. (2022) when inflammatory online rhetoric corresponded with documented increases in physical aggression toward Asian communities (Han, Riddell & Piquero, 2023). Given these serious implications, including the documented correlation between hate speech and escalating violence against minority populations (Laub, 2019), the development and implementation of sophisticated computational systems for identifying and moderating discriminatory content has become a critical priority for digital platform governance.

The academic community has produced extensive research addressing digital hate speech detection, yielding various methodological approaches and technological solutions (Schmidt & Wiegand, 2017; Del Vigna et al., 2017). Contemporary detection systems, primarily utilizing transformer architectures and advanced neural networks (Sheth et al., 2023), achieve notable accuracy metrics in standardized testing environments. However, these sophisticated computational models function as black boxes, offering minimal insight into their executive processes. This opacity becomes particularly problematic amid hate speech identification, where algorithmic transparency is not merely beneficial but essential. Davidson et al. (2017) research has demonstrated that classification errors can paradoxically reinforce discriminatory patterns against the very demographics. These systems aim to protect (Sap et al., 2019). Consequently, developing interpretable models serves dual purposes: enabling users to comprehend automated decisions and facilitating the identification of systematic biases and algorithmic shortcomings.

Current approaches to algorithmic transparency encompass various analytical frameworks, with two prominent methodologies emerging in recent literature. The SHapley Additive exPlanations (SHAP) methodology (Lundberg & Lee, 2017) quantifies the relative contribution of individual variables to specific model outputs through a game-theoretic framework. Complementing this approach, local interpretable model-agnostic explanations (LIME) (Ribeiro, Singh & Guestrin, 2016) enhances model transparency by constructing simplified, interpretable approximations of complex decision boundaries in the vicinity of individual predictions. Nevertheless, these analytical tools present significant computational challenges when applied across large datasets. Furthermore, research indicates an inherent tension between model complexity and interpretability (Dziugaite, Ben-David & Roy, 2020), particularly in sophisticated architectures. The nuanced nature of potentially discriminatory language necessitates contextual analysis comparable to human cognitive processing. As Kim, Lee & Sohn (2022) argue, effective hate speech detection systems must provide contextually grounded explanations accessible to human reviewers. While integrating interpretability mechanisms directly into neural architectures remains technically challenging, an alternative framework involves developing supplementary models dedicated to generating explanatory rationales. These supporting systems can then inform the training process of the primary detection algorithm, creating a more transparent classification process.

A pioneering approach to algorithmic transparency was introduced through the Faithful Rationale Extraction from Saliency tHresholding (FRESH) methodology (Jain et al., 2020), which implements a dual-network architecture: one component identifies task-relevant textual elements, while a separate network utilizes these elements for classification purposes, establishing interpretability as a fundamental design feature. While FRESH demonstrated the viability of this approach through simplified architectural design and token-based feature selection, its explanatory capacity remains restricted to the isolated textual components identified during processing. Our research extends beyond these limitations by incorporating advanced language models (LLMs) as sophisticated feature extraction mechanisms in hate speech identification systems. This novel framework capitalizes on the semantic processing capabilities and directive responsiveness characteristic of contemporary LLMs to derive contextually relevant textual indicators. These extracted elements subsequently enhance the training process of a dedicated hate speech classification system, yielding an inherently interpretable methodology. This study is guided by the following key research questions: 1. RQ1: To what extent can Mistral-7B contribute effectively to the task of hate speech detection across our experimental datasets?

2. RQ2: Can recent state-of-the-art LLMs be leveraged to extract rationales as meaningful features, and can these rationales potentially replace human annotations?

3. RQ3: To what extent can hallucinations in LLMs impact the process of hate speech detection, and what strategy can effectively reduce these hallucinations?

4. RQ4: Can TARGE enhance the performance of the hate speech detector while also offering transparent, reliable explanations that reflect its decision-making process?

Based on these research questions, our study makes the following significant contributions: A novel framework, TARGE, is introduced, utilizing rationales generated by large language models (LLMs) to enhance a base model for detecting hate speech, ensuring both interpretability and fidelity. This minimizes the need for task-specific fine-tuning and extensive human annotation.

By incorporating LLM-extracted rationales into the base hate speech detector, we ensure explanations are inherently aligned with the model’s reasoning, thus achieving faithful explainability without compromising detection performance.

Our methodology innovatively combines the base detector’s [CLS] embedding with a separate embedding of the LLM-extracted rationales. This concatenated embedding strategy leverages both the holistic context of the input text and the targeted, interpretable features, resulting in improved detection performance-especially evident in noisy data scenarios like the Twitter dataset.

Introduces an iterative framework that uses a score-refine strategy, enabling LLMs to assess and correct hallucinated content.

TARGE’s outcomes are interpreted using Integrated Gradients from Captum, showcasing the framework’s capability to deliver understandable insights into its hate speech detection decisions.

Literature review

The identification and regulation of discriminatory discourse represents a critical challenge in digital communications research, requiring sophisticated methodologies that protect community standards while preserving legitimate expression. In contemporary society, where social media platforms significantly influence public discourse, developing effective mechanisms to counteract the societal impact of inflammatory rhetoric has become paramount.

Traditional approaches in hate speech detection

Although current computational approaches demonstrate considerable efficacy in detecting problematic content, their architectural complexity often obscures the underlying analytical process. Contemporary machine learning systems, despite their accuracy, frequently operate through opaque computational processes that resist straightforward analysis. The development of transparent classification systems would serve multiple objectives: enhancing user confidence through algorithmic accountability, facilitating deeper technical understanding of detection mechanisms, and ultimately enabling the creation of more sophisticated content moderation frameworks that effectively balance social responsibility with expressive freedom. The escalating significance of discriminatory content moderation in digital spaces has emerged as a focal point of computational linguistics research. Academic investigation into automated detection systems has produced diverse methodological frameworks, each addressing distinct aspects of online discourse analysis and content classification. The following literature review examines seminal contributions to this evolving field, synthesizing crucial developments in algorithmic approaches to inflammatory speech identification. Initial research efforts employed traditional statistical learning approaches for automated content classification, as exemplified by Davidson et al. (2017), which introduced an extensive annotated corpus and implemented classical algorithms—including logistic regression (Joachims, 1998) and support vector machines (SVM) (Wright, 1995)—using n-gram feature extraction. Early studies on hate speech detection similarly relied on conventional machine learning techniques such as SVM, k-nearest neighbors (k-NN), random forest, and decision tree models that leveraged diverse feature representations (e.g., syntactic structures, semantic information, sentiment analysis, and lexical attributes) (Mullah & Zainon, 2021). Although these classical approaches effectively captured lexical patterns, they demonstrated inherent limitations in processing the contextual and semantic relationships crucial for accurately identifying inflammatory content—a gap that later motivated the exploration of deep neural networks (Sun et al., 2021). Recurrent neural networks (RNNs) and convolutional neural networks (CNNs) have manifested as leading methods for hate speech detection. The selection of deep learning design frequently depends on the characteristics of the textual data under analysis. For instance, CNNs are frequently employed for shorter texts, where capturing intricate contextual information is less critical. Their ability to effectively identify local patterns has made them a preferred choice for various text classification applications (Wang et al., 2020; Zhou et al., 2022; Xu et al., 2020).

On the other hand, for extended text sequences that require thorough insights into semantic features and contextual relationships, RNNs, especially long short-term memory (LSTM) networks and bidirectional LSTMs (BiLSTMs) tend to outperform other methods (Du, Vong & Chen, 2021; Sari, Rini & Malik, 2019; Shi, Wang & Li, 2019).

Warner & Hirschberg (2012) carried out an early and groundbreaking study in hate speech detection, emphasizing the identification of anti-Semitic expressions as a unique category within this domain.

Waseem & Hovy (2016) performed an important finding focusing on hate speech detection on Twitter, specifically addressing instances of racism and sexism. Their research examined various features, including user demographic attributes, lexical patterns, geographic data, and character-level n-grams. Among these, character n-grams of up to four characters were identified as the most effective for the task. Additionally, incorporating gender as a supplementary feature resulted in a modest enhancement of the classification performance.

Khan et al. (2022) presented BiLSTM with deep CNN and Hierarchical Attention-based deep learning model for tweet representation (BiCHAT), a neural network architecture combining Bidirectional Encoder Representations from Transformers (BERT)-based embeddings with BiLSTM and deep convolutional layers. It incorporates a multi-level attention mechanism that functions at both word and sentence levels, allowing the model to focus on critical words and phrases while filtering out less pertinent details. The effectiveness of the proposed framework was validated on the widely-used Twitter hate speech dataset, where it demonstrated superior performance compared to the baseline model.

Kapil & Ekbal (2020) proposed a multi-task learning model developed to identify different but related types of hate speech, such as racism, offensive language, and sexism. Their methodology included various neural architectures, such as CNNs, LSTM networks, and a hybrid architecture merging CNNs with gated recurrent units (GRUs).

Fortuna, Soler-Company & Wanner (2021) proposed an in-depth study of the classification of hate speech, abusive language, toxicity, and offensive content. Their work explored different models, including BERT, A Lite BERT (ALBERT), fastText, and SVM, using nine publicly available datasets. The research evaluated model performance both within individual datasets and across multiple datasets, assessing their ability to generalize across different hate speech categories and data distributions.

Hate speech detection progresses at a rapid pace, fueled by progress in machine learning and multimodal methodologies. While substantial headway has been made, key challenges remain, such as reducing biases and strengthening defenses against adversarial intrusions. Addressing these challenges is necessary for developing more reliable and effective detection systems, and fostering safer and more inclusive digital environments.

Explainable approaches in hate speech detection

In response to the black-box nature of deep models, several researchers have explored methods for making hate speech detectors more interpretable. For instance, Calabrese et al. (2024) introduced structured explanation techniques that highlight harmful spans within a post to assist human moderators in making faster and more accurate decisions. Their work demonstrated that structured, post-specific explanations can reduce moderation time, yet their primary focus was on enhancing human efficiency rather than embedding interpretability directly into the model’s prediction process.

LLM-based techniques

Recent studies highlight the versatility of large language models, demonstrating their effectiveness in applications such as data annotation (Bhat & Varma, 2023; He et al., 2024), text classification (Bhattacharjee & Liu, 2024; Kocoń et al., 2023), and reasoning (Wang et al., 2024). Recent investigations into the behavior of large language models in the context of hate speech detection have revealed that these models can exhibit excessive sensitivity. Zhang et al. (2024) highlight how some LLMs tend to misclassify benign content as hateful due to over-sensitivity toward certain groups or topics, and they also note challenges in confidence calibration. Although this evaluation is critical for understanding the limitations of LLMs, it primarily serves as a cautionary tale regarding their direct application in detection tasks rather than offering a solution to enhance interpretability. Parallel to the work on detection, another stream of research has concentrated on generating counterspeech responses that help mitigate hate speech. Hong et al. (2024) proposed outcome-constrained large language models that generate counterspeech designed to steer conversations toward lower incivility or encourage non-hateful reentry. Although this research leverages the capabilities of LLMs to produce linguistically and contextually nuanced responses, its focus remains on reply generation rather than on explaining the underlying classification decisions. In other words, while these models excel at influencing conversation outcomes, they do not necessarily improve the transparency of hate speech detection systems.

Zero- and few-shot learning techniques

Hate speech detection has experienced notable advancements through data-efficient strategies such as zero-shot learning (ZSL). One prominent research direction employs prompting techniques with instruction fine-tuned language models. For instance, Plaza-del arco & Hovy (2023) illustrate that carefully designed prompts and verbalizers—such as the “respectful-toxic” pair—can yield competitive performance across diverse datasets and languages. However, while this ZSL approach advances detection accuracy, it tends to operate as a black box, offering limited insight into the model’s internal decision-making process. In another line of inquiry, Yuzbashyan et al. (2023) proposes a zero-shot method that reframes hate speech detection as a natural language inference (NLI) task. In their approach, a hypothesis (e.g., “This text is racist”) is paired with a target sentence, and an NLI model evaluates whether the hypothesis is entailed by the text. Their experiments, conducted over multiple datasets, reveal that although this NLI-based zero-shot method can rival supervised learning approaches, its performance is highly sensitive to the exact phrasing of the hypothesis; even minor lexical variations can lead to substantial fluctuations in the F1-score, raising concerns about its robustness and generalizability. A further advancement in this domain is presented by Goldzycher & Schneider (2022), who explore hypothesis engineering for zero-shot hate speech detection. Their work repurposes NLI models by formulating a range of hypotheses (e.g., “That contains hate speech.”) and combining multiple supporting hypotheses to mitigate common errors, such as the misclassification of counterspeech, reclaimed slurs, or dehumanizing comparisons. While this hypothesis engineering strategy enhances performance in low-resource settings, it relies heavily on the manual formulation and meticulous selection of hypotheses. This reliance not only increases computational overhead—given the need for multiple forward passes per hypothesis— but also reduces transparency in understanding how decisions are ultimately made. In contrast to ZSL methods, TARGE integrates LLM-extracted rationales into its detection process. This not only maintains competitive accuracy but also provides transparent, interpretable insights that enhance trust and facilitate error analysis.

System model

Preliminary

The advent of social media has transformed communication and self-expression, creating a virtual space for individuals to engage in dialogue and share their perspectives. However, the obscurity and assumed absence of responsibility on these platforms have contributed to the spread of offensive and hate speech content (Ullmann & Tomalin, 2020). With the expanding influence and widespread use of these platforms, the development of automated systems to detect and address hate speech has become an essential priority. Various approaches to hate speech detection have been proposed, yet many depend on intricate deep learning models that function as black-box systems with limited clarity and explainability (Guidotti et al., 2018). Interpretability, which refers to the ability of humans to understand the reasoning behind a decision (Miller, 2019), remains a critical but frequently neglected aspect in these models. This deficiency raises significant concerns regarding potential biases and inaccuracies in predictions. Ensuring interpretability in hate speech detection systems is essential to fostering user trust, enhancing the understanding of decision-making processes, and enabling the development of more equitable and reliable solutions (Felzmann et al., 2020). LLMs have revolutionized artificial intelligence (AI) research by showcasing exceptional proficiency in generating contextually rich text and managing intricate tasks (Hadi et al., 2023). In the realm of misinformation detection, LLMs are being employed to create more robust systems for identifying fake news, particularly targeting disinformation produced by LLMs. Furthermore, their proficiency in natural language tasks, such as stance detection, has shown results comparable to human annotations, prompting researchers to explore their potential in automating annotation processes. Building on this approach, our goal is to harness LLMs to automate the extraction of rationales from human annotations tailored to our use case. By using LLMs in a one-shot setting, we aim to produce superior rationales while reducing the biases typically associated with these models. This method leverages the sophisticated language understanding and generation abilities of LLMs to ensure both reliable predictions and interpretable outcomes. Evaluations using a comprehensive social media-rich dataset, which incorporates text from multiple social media platforms, validate our framework’s effectiveness in two critical areas: The quality and alignment of rationales extracted by LLMs.

The ability to retain detector performance while incorporating interpretability, challenging the assumed trade-off between accuracy and interoperability.

Rationale extraction

Our framework utilizes advanced instruction-tuned LLMs as pre-trained tools for extracting textual features. While prior research indicates that LLMs underperform in hate speech detection without fine-tuning or auxiliary models (Li et al., 2023; Zhu et al., 2023), we propose leveraging their language comprehension capabilities to extract rationales as textual features. By confining LLMs to text-level tasks, we avoid directly applying them to sensitive domains like hate speech detection, addressing concerns about bias and limitations (Harrer, 2023).

This approach strategically employs LLMs as auxiliary feature extractors, capitalizing on their strengths in text analysis while assigning hate speech detection to a specialized model. By separating feature extraction from classification, we balance the advantages of LLMs with the need for reliable and unbiased detection systems. The feature extraction process involves prompting the LLM with a specific query for each input text, as shown in Fig. 1. Although LLMs demonstrate impressive potential and have advanced significantly, they still encounter a critical issue known as “hallucination,” where they produce responses that sound plausible yet are ultimately inaccurate or nonsensical. To assess how hallucinations in LLMs may affect hate speech detection and to validate the interpretability of our results with real human judgment, we compare them against human-annotated rationales from a reference dataset (HateXplain) using similarity metrics. At the token level, we compute overlap similarity to assess the degree of textual correspondence between the generated and human rationales. In addition, we calculate cosine similarity in the latent space—by encoding the texts with the Universal Sentence Encoder—to capture semantic alignment. For our experiments, we set a threshold of 0.50 for the token-level metrics and 0.70 for the cosine similarity. If the scores fall below these thresholds, the generated rationale is deemed hallucinated or unreliable.

Figure 1 Task prompt.

In such cases, the system automatically re-prompts the language model with modified instructions as shown in Fig. 2 to produce a refined rationale. This iterative score-refine loop continues until the refined rationale satisfies the similarity criteria, thereby ensuring that it is factually grounded and closely aligns with human judgment.

Figure 2 Task re-prompt.

The rationales and extracted features act as additional inputs for a tailored hate speech detection model, improving its capacity to provide more accurate and transparent predictions. This approach capitalizes on the LLM’s proficiency in text analysis while assigning the critical process of hate speech classification to a tailored model.

We compute the similarity between the Mistral-7B-extracted rationales for the input text from the HateXplain dataset and the corresponding human-annotated rationales using our defined similarity metrics. The resulting scores, which represent the baseline performance prior to any intervention, are presented in Table 1. Subsequently, Table 2 shows the similarity metrics after our automated hallucination reduction process has been applied.

Table 1 Similarity between HateXplain’s human-annotated explanations and Mistral-7B rationales before hallucination removal.

Similarity metric	Similarity coefficients (%)	
Overlap similarity	90.50	
Cosine similarity	69.00	

Table 2 Similarity between HateXplain’s human-annotated explanations and Mistral-7B rationales after hallucination removal.

Similarity metric	Similarity coefficients (%)	
Overlap similarity	94.85	
Cosine similarity	72.30	

Embedding module

The next crucial element of our framework is the core hate speech detection model, implemented using DistilBERT. DistilBERT, a lighter and more efficient variant of the BERT model (Devlin et al., 2019), is trained on an extensive dataset to preserve BERT’s essential functionalities while enhancing efficiency. For our application, rather than focusing on output labels or class probabilities for an input text ti∈T, we extract the embedding from the final layer of the CLS token, h[CLS]i. This embedding captures the most critical semantic and contextual information extracted from the input text, specifically tailored for hate speech detection.

Utilizing the pre-trained and fine-tuned embeddings offered by DistilBERT, the framework gains a concise yet rich encoding of the input data. Rather than solely depending on the final classification result, the [CLS] token embedding acts as a dense representation of the input’s features and semantics. This approach strengthens the base detector by integrating supplementary features and rationales derived from the large language model, resulting in a more robust and interpretable system for detecting hate speech.

Feature embeddings

After processing the outputs, we extract a set of s textual features, denoted as z1,z2,…,zs, from the input text ti. To embed these features and rationales generated by the LLM, we utilize a pre-trained transformer-based language model RoBERTa. This model, even without task-specific fine-tuning, generates comprehensive and expressive latent representations of text. Specifically, the LLM-extracted textual features are fed into the RoBERTa-base model, and the embedding corresponding to the [CLS] token in its final hidden layer, represented as hCLS,ift, is obtained.

By leveraging a pre-trained model such as Robustly Optimized BERT Pretraining Approach (RoBERTa), we produce embeddings that are both semantically rich and contextually informed. These embeddings, hCLS,ift, encapsulate the semantic and contextual essence of the LLM-derived features and rationales. Their integration into our hate speech detection framework enhances the base detector by incorporating valuable complementary insights provided by the LLM-generated outputs. The robust representations from RoBERTa, even without task-specific fine-tuning, allow for a more comprehensive and effective detection system.

Fusion and classification

For each input text ti, two embeddings are obtained from the prior components: the text embedding Etext,i derived from the core hate speech detection model and the embedded features Efeat,i generated by the feature embedding model based on RoBERTa. To integrate these embeddings, we concatenate them as follows:

Ecombined,i=Etext,i⊕Efeat,i.

This combination integrates the task-specific representation from the core detector with the contextual features and rationales obtained from the LLM-extracted textual elements, creating a unite representation. Ecombined,i. This comprehensive embedding captures complementary information, enhancing its utility for the final hate speech classification task.

The concatenation process facilitates a smooth integration of the two embeddings, maintaining their distinct contributions while allowing the final classifier to utilize the merged representation efficiently. By blending these diverse features, the pipeline leverages the strengths of both the core detector and the enriched textual insights provided by the LLM, enhancing both interpretability and decision-making capabilities.

Unlike previous studies, which relied solely on extracted rationales for downstream tasks, our approach combines them with additional contextual embeddings, providing a richer input. The concatenated representation Ecombined,i is input into a feed-forward multi-layer perceptron (MLP) composed of two fully connected layers with a rectifier linear unit (ReLU) activation function (Agarap, 2018) in between. This MLP projects the combined embedding onto a lower-dimensional space to retain essential features while reducing overfitting during training. Following prior methodologies (Pan et al., 2022) this projection ensures robust feature utilization.

The training aim is to minimize the batch-wise binary cross-entropy loss. For a batch size of n, the loss is computed as:

LossCE=−1n∑i=1n[yilog⁡(p(yi|f(Ecombined,i)))+(1−yi)log⁡(1−p(yi|f(Ecombined,i)))].

Here, yi represents the ground truth label for the input xi, and f(⋅) denotes the MLP that processes the concatenated embedding. The RoBERTa feature embedding model remains frozen during training, ensuring it only serves as a contextual encoder for the extracted textual features z.

This approach generates a comprehensive representation by merging embeddings from the core detector with features derived from the LLM. The resulting concatenated embedding provides a valuable input for the final classifier, enabling it to utilize complementary insights for improved prediction accuracy. Employing an MLP for dimensionality reduction helps preserve essential features while reducing the risk of overfitting, thereby increasing the model’s overall robustness and ability to generalize effectively.

Experiments

To execute the proposed TARGE framework, we utilized PyTorch in combination with the Hugging Face Transformers library, as illustrated in Fig. 3. The initial phase of the framework leverages a pre-trained LLM to extract features and rationales. For this step, we employed Mistral-7B, recognized for its superior performance on multiple NLP tasks (Jiang et al., 2023). Mistral-7B was selected for its strong performance in instruction-following tasks and computational efficiency, providing an optimal balance between model size, speed, and accuracy suitable for rationale extraction in resource-constrained environments. Within the framework, a pre-trained and frozen RoBERTa model (roberta-base) serves as the Feature Embedding Model, while a pre-trained DistilBERT model functions as the hate speech detector. To facilitate efficient training, the AdamW optimizer is employed with a learning rate of 2×10−5. All experiments are assessed using accuracy as the primary performance metric, ensuring consistency and reliability in the evaluation of results. To enhance robustness and detection performance, we employed a heterogeneous embedding approach where DistilBERT encodes the original offending message, and RoBERTa encodes the rationale text generated by Mistral-7B. DistilBERT’s distilled architecture allows efficient processing of raw text with reduced computational overhead, making it well-suited for encoding offensive messages. Conversely, RoBERTa, with its robust pre-training and superior contextual representation capabilities, effectively captures complex semantic nuances from the generated rationales. This complementary embedding strategy introduces diversity in the feature space, enhancing the model’s ability to capture a broader range of linguistic and semantic cues. Additionally, the integrated gradients (IG) method (Sundararajan, Taly & Yan, 2017) as implemented in the Captum library was employed to explain the predictions of the hate speech detection model.

Figure 3 Proposed framework architecture.

Datasets

To analyze the performance of the proposed TARGE framework, we employed the ETHOS dataset (Mollas et al., 2022). The dataset is publicly available at the official GitHub repository: https://github.com/intelligence-csd-auth-gr/Ethos-Hate-Speech-Dataset/tree/master/ethos/ethos_data. The ETHOS dataset is a well-curated collection of hate speech data sourced from diverse social media platforms, including YouTube and Reddit. This English-language dataset is annotated in detail, making it suitable for binary and multi-label classification tasks. For binary classification, the dataset includes 998 comments labeled to indicate whether hate speech is present or absent. For multi-label classification, the Ethos Multi-Label subset consists of 433 instances of hate speech, annotated across multiple categories, including violence, gender, race, national origin, disability, sexual orientation, and religion. The ETHOS dataset was constructed through data collection efforts on the Hatebusters platform and Reddit’s publicly available repositories. The annotation process was further validated via the Figure-Eight platform, ensuring both reliability and diversity. These rigorous steps establish ETHOS as a benchmark dataset, offering a robust foundation for evaluating hate speech detection systems. In our experimental setup on ETHOS dataset, we reserved 12.5% of the total data exclusively for testing, which is used only for final evaluation. Of the remaining data, 75% was allocated for training while the final 12.5% served as a validation set. This split ensures that the model is robustly trained and hyperparameters are effectively tuned prior to final evaluation. Our study uses the Mathew et al. (2021) HateXplain dataset as another benchmark for explainable hate speech detection. HateXplain is a pioneering dataset that not only provides the traditional three-class labels—hate speech, offensive speech, and normal—but also incorporates additional layers of annotation that are crucial for explainability. The dataset comprises approximately 20,148 posts collected from two prominent social media platforms: Twitter (9,055 posts) and Gab (11,093 posts). This dual-source collection ensures a diverse representation of hate speech and provides insights into platform-specific language usage and context. For our experiments, we adopted a two-phase evaluation approach: Split-version evaluation: Initially, we conducted separate experiments on the Twitter and Gab sub-datasets as shown in Table 3. This allowed us to analyze the characteristics and performance of our models on platform-specific data, understanding nuances that might arise from the distinct nature of each source. Combined-version evaluation: Subsequently, we used a unified data set. This combined version was employed to benchmark our results against the evaluation framework proposed in the original HateXplain work, enabling a comprehensive comparison of performance as shown in Table 4.

Table 3 Results for our TARGE framework (highlighted in bold) vs. The baseline models.

Model	F1-score	Accuracy	Precision	Dataset	
BiLSTM + static BE (Rajput et al., 2021)	79.71	80.15	80.37	Ethos	
BERT (Mollas et al., 2022)	78.83	76.64	79.17	Ethos	
BiLSTM + Attn FT (Mollas et al., 2022)	76.8	77.34	77.76	Ethos	
DistilBERT (Mollas et al., 2022)	79.92	80.36	80.28	Ethos	
SVM (Mollas et al., 2022)	66.07	66.43	66.47	Ethos	
Random Forests (Mollas et al., 2022)	64.41	65.04	64.69	Ethos	
TARGE (Proposed)	82.01	87.05	82.04	Ethos	
DistilBERT	77.02	80.64	78.47	GAB	
Mistral-7B-1shot	81.31	81.03	80.86	GAB	
TARGE (Proposed)	90.32	91.26	90.85	GAB	
DistilBERT	51.91	52.26	50.73	Twitter	
Mistral-7B-1shot	55.21	56.07	54.75	Twitter	
TARGE (Proposed)	63.02	62.83	62.15	Twitter	

Table 4 Performance comparison of TARGE FRamework (highlighted in bold) vs. Non-LLM baseline model.

Model	F1-score	Accuracy	Dataset	
BERT-HateXplain [Attn] (Mathew et al., 2021)	0.687	0.698	HateXplain	
TARGE (Proposed)	0.766	0.770	HateXplain	

Results

This section presents a thorough explanation of the experiments conducted and an in-depth examination of the results to evaluate the practicality and effectiveness of the proposed TARGE framework. The investigation assesses whether the TARGE framework can sustain or enhance the performance of the hate speech detection system while delivering faithful interpretability. This evaluation addresses the critical trade-off between achieving high predictive performance and maintaining model transparency, aiming to meet both objectives effectively. Table 5 presents the faithfulness metrics for explainability, which include comprehensiveness and sufficiency, to evaluate how well the integrated gradients-based explanations capture the decision-making process of our model. Comprehensiveness is measured by removing the highly attributed words from the input and quantifying the drop in the model’s predicted probability for the target class; a larger drop indicates that these words are critical for the decision. Conversely, sufficiency is determined by providing only the influential words and comparing the resulting prediction to that obtained with the full input. A small difference here implies that the selected words are sufficient to preserve the original prediction, effectively capturing the essential factors behind the model’s decision. Together, these metrics offer a rigorous assessment of the model’s explanation fidelity. The TARGE framework enhances interpretability by integrating extracted rationales into the input while maintaining high accuracy. Figures 4–6 illustrate the key influential words identified by the model using integrated gradients.

Table 5 Faithfulness metrics for explainability—selected for the TARGE model.

Model	Comprehensiveness	Sufficiency	
TARGE	0.74	0.001	

Figure 4 Integrated gradients (IG) visualization of the proposed framework’s performance on the GAB dataset.

Figure 5 Integrated gradients (IG) visualization of the proposed framework’s performance on the Twitter dataset.

Figure 6 Integrated gradients (IG) visualization of the proposed framework’s performance on the ETHOS dataset.

LLM performance evaluation

This study examines the capability of Mistral-7B to understand text and context, with a specific focus on extracting features pertinent to hate speech detection. Mistral-7B-v0.1, a state-of-the-art LLM, is utilized as the feature extraction component, leveraging the advanced instruction-following abilities characteristic of modern LLMs. A carefully crafted prompt (illustrated in Fig. 1) facilitates the extraction of rationales, offensive language, and profanities from the input text. These extracted features are subsequently provided as interpretable inputs to the predictor model, DistilBERT, ensuring a transparent and dependable interpretation of hate speech detection outcomes. To build on prior research, we designed a one-shot prompt that guides Mistral-7B to classify a given text using a single labeled example. This prompt returns a binary result, assigning a “1” to texts identified as hateful and a “0” to those deemed non-hateful, as depicted in Fig. 7. We classify the data across two datasets GAB and Twitter and measure the resulting accuracy. The performance of this one-shot classification approach is then compared with that of the baseline models, and the outcomes are presented in Table 3. We observe a clear contrast between the baseline models and Mistral-7B’s one-shot classification performance. Although this indicates that LLMs—may not excel as standalone hate speech detectors, their strong capabilities in understanding textual nuances remain impressive.

Figure 7 Mistral-7B one-shot hate speech detection prompt and response.

Hate speech detector performance

This experiment aims to improve the interpretability of hate speech detection by incorporating extracted rationales into the input text during model training. DistilBERT is utilized as the base model for hate speech detection, with the results presented in Table 3, alongside comparisons with other baseline methods. The findings indicate that the proposed TARGE framework achieves performance comparable to the fine-tuned DistilBERT model on the same dataset. This retention of performance is noteworthy, as interpretability-focused models often sacrifice accuracy (Dziugaite, Ben-David & Roy, 2020; Bertsimas et al., 2019).

Limitations

While our iterative-refinement method demonstrates effectiveness in reducing hallucinations through similarity-based comparisons with human-annotated rationales, several aspects require further attention. A notable consideration is the method’s dependence on the availability of expert-provided annotations, which may not always be feasible for completely unseen texts. Future research will therefore explore unsupervised consistency checks and annotation-free approaches to further enhance the robustness and generalizability of our hallucination reduction method.

Conclusion

In this work, we demonstrate that although Mistral-7B is not competitive as a standalone zero-shot hate speech detector, it is highly effective in generating high-quality rationales. When these rationales are integrated through our proposed TARGE framework, the resulting model achieves classification performance comparable to that of a strong supervised baseline. By training exclusively on LLM-generated rationales, we show that machine-derived explanations can serve as effective supervisory signals, achieving interpretability and decision consistency comparable to models trained with human annotations. Furthermore, we address the challenge of hallucinated rationales by introducing a similarity-based filtering strategy, which effectively removes spurious spans without compromising recall, thereby enhancing the reliability of the model’s explanations. Overall, TARGE successfully combines these advancements into a unified framework that maintains high predictive accuracy while offering transparent, token-level justifications for each prediction. This work provides a promising direction for developing interpretable and trustworthy hate speech detection systems for social media platforms.

The authors acknowledge the use of OpenAI’s ChatGPT for proofreading and editing the manuscript to improve its clarity, grammar, and coherence.

Additional Information and Declarations

Competing Interests

The authors declare that they have no competing interests.

Author Contributions

Muhammad Haseeb Hashir conceived and designed the experiments, performed the experiments, performed the computation work, prepared figures and/or tables, and approved the final draft.

Memoona conceived and designed the experiments, analyzed the data, prepared figures and/or tables, authored or reviewed drafts of the article, and approved the final draft.

Sung Won Kim analyzed the data, authored or reviewed drafts of the article, and approved the final draft.

Data Availability

The following information was supplied regarding data availability:

The Ethos-Hate-Speech-Dataset is available at GitHub:

https://github.com/intelligence-csd-auth-gr/Ethos-Hate-Speech-Dataset/tree/master/ethos/ethos_data.

The code is available at GitHub and Zenodo:

- https://github.com/Haseeb-29/peerj

- Haseeb-29. (2025). Haseeb-29/peerj: V2 (Version V2). Zenodo. https://doi.org/10.5281/zenodo.15170401.

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
