# Peer review of "TARGE: large language model-powered explainable hate speech detection"

_PeerJ Computer Science, doi:10.7717/peerj-cs.2911_

## Round 0.1 · original submission · Major Revisions

Both reviewers recommend the paper to undergo major revisions to address their concerns before it can be considered for review again.

Reviewer 1 ·

Basic reporting

The article presents a system for detecting offensive messages on social media through a framework called TARGE. This framework utilizes Mistral 7B for feature extraction, a frozen RoBERTa model as the Feature Embedding Model, and DistilBERT for hate speech detection. By leveraging Mistral’s language compression capabilities, the detection system becomes more explainable and follows a logical reasoning process, improving overall performance. While the study is interesting, I have several comments for consideration:

1. Typographical Issues:
- The article contains some typos that affect readability. For example:
Missing periods on line 39 (after “inherent characteristics” and before “Including, but not”), line 55 (after “against very demographics” and before “these systems”), and line 205 (after “unbiased detection systems” and before “The feature”).
- Incorrect capitalization in multiple instances, such as line 54 (“Davidson et al. (2017) Research”), line 185 (“Building on this approach, Our”), and line 219 (“solely depending On the”).
Some references do not display the publication year correctly (e.g., line 67 “Kim et al. (022a)” and line 198 “Zhu et al. (023b)”).
- The sentence in line 115, which includes references, is difficult to read. It would be clearer if the references were placed within parentheses.
- The paragraph following line 223 contains CLS as a standalone line, which should be merged with the surrounding text.
- References on lines 181 and 182 should be enclosed in parentheses.

2. Acronyms in the Title and Abstract:
- The acronym LLM should not appear in the title. Additionally, the abstract does not define what an LLM is.
- The abstract should be more specific and aligned with the research, clearly stating the problem addressed, the methodology used, and the results obtained.

3. The bibliographic review does not thoroughly discuss existing offensive language detection systems that have emerged since the introduction of LLMs. Many of these systems closely resemble the one proposed in this study.

4. The first two sections do not indicate which language the research applies to. This information should be explicitly mentioned.

5. Paragraphs 111–118 and 119–126 describe very similar language detection systems using classical methodologies. These sections could be merged into a single, more concise paragraph summarizing their content.

6. Figure 3 appears too small. It may be beneficial to split it into two subfigures and enlarge each for better readability.

Experimental design

The experimental design appears logical. Moreover, it follows a strategy that is just beginning to be applied to leverage the reasoning capabilities of generative language models. Additionally, Figure 2 provides a visual representation that helps clarify the workflow. I have a few comments regarding this:

7. It is not entirely clear why Mistral was chosen as the LLM instead of other more well-known models such as LLaMA or GPT. Similarly, the rationale behind using DistilBERT to extract the embeddings of the offending message and RoBERTa for the embeddings of the text generated by Mistral is unclear. Intuitively, it would seem more appropriate to use the same model for generating embeddings and encoding the language. Did the authors intend to introduce greater diversity or more varied embeddings by using two different models?

8. The paper does not mention the amount of data used to train the system or the data partitions applied for this purpose, despite Figure 4 displaying results per epoch, suggesting that this methodology was followed.

9. The paper lacks an error analysis, which would help to understand the system’s strengths and weaknesses. Specifically, it would be useful to determine whether errors stem from flawed reasoning by the LLM or from misclassification by the classifier.

Validity of the findings

I find the research interesting; however, the methodology is already being applied, and there are published articles on this topic in major NLP conferences. Therefore, the authors should emphasize the novelty of their work compared to these previously published studies, which are not even referenced in the literature review. Additionally, the review should mention other methods based on ZSL or FSL to help illustrate how the proposed approach improves upon existing systems (as is done in some related works).

Furthermore, this study clearly demonstrates that, for the same dataset, the proposed system outperforms previous approaches. The conclusions are well-reasoned and clearly presented.

Additional comments

No additional comments

Cite this review as

·

Basic reporting

1. The related work section can be better organized. There's no particular order currently. I recommend splitting it into two subsections: hate speech detection & explainability.
2. The RNN & CNN section in the related work section seems out of the blue. They are also outdated compared to transformers.
3. Line 98-101. These two sentences read quite redundant.
4. The abstract and the introduction are clearly written, but the rest of the paper isn't. It's clear that the advisor didn't revise the whole manuscript carefully before submitting.

Experimental design

1. To prove the reliability of the method, should consider reporting results on at least 3 datasets. However, only 1 is provided.
2. All the baselines except for DistilBERT are not transformer-based, which is the current status quo.
3. They mentioned the bias of LLMs. However, didn't report the performance of the LLM. Can consider prompt the LLM to generate the predicted label alongside the extracted span.
4. I'm surprised there's no quantitative evaluation of the generated explanations.
5. Line 77: The authors pointed out the limitations of Jain et al. 2020. But they didn't compare with it to show that the proposed method is superior.
6. LIME is a very old explainability method. Should adopt more recent gradient or activation based methods that are more suited for DNNs.
7. I believe multi-task learning of detecting hateful span and classification isn't new. You can consider compare with this baseline that doesn't use LLM, but the base model to extract the span.

Validity of the findings

Sufficient details are provided and the model should be fairly easy to reproduce.

Additional comments

1. Line 111: What's the difference between "computational analysis of discriminator discourse" and "Hate Speech Detection"? The paper Davidson et al. 2017 is actually about the latter but you put it under the earlier.
2. Line 200: I don't understand how you address the bias problem of LLMs. What evidence supports this claim?

---

## Round 0.2 · Minor Revisions

Please address the final minor revisions raised by both reviewers before the paper can be accepted for publication.

Reviewer 1 ·

Basic reporting

In this new version, the text is written more clearly. There are no more sentences with typing errors, too long or difficult to read. In addition, the summary shows the main objectives of the research carried out, the methodology and the results obtained.

Another aspect to value is the inclusion of the research questions that clarify everything a little more.

However, there are some small adjustments that still need to be made:
- On line 198 there appear 4 references marked as ‘?’.
- On lines 279, 447 and 448 there is text between quotation marks in which the first inverted commas are misspelled.
- In the text of the article, the Mistral model used is sometimes referred to as ‘Mistral 7B’ and sometimes as ‘Mistral-7B’.
- In Figures 1 and 7 there is a lot of white space within the image boxes.
- Finally I would be more careful with the 1 column formatting of the article as in lines 380 and 383 some of the text appears in the margins of the article.
- In the caption of the tables it should be indicated that the text is in bold in order to make it clearer.

Experimental design

I am grateful to the authors for including more clarity on why they selected these models, as well as including new experiments that have enriched their research.

Validity of the findings

More recent research has been looked for that justifies the reason for the current work. However, there is one aspect that could be improved for the final article. Since a series of research questions are posed in the introduction to the article, I would respond to these questions in the conclusions by referring to them.

Additional comments

No additional comments.

Cite this review as

·

Basic reporting

The authors reorganized their manuscript to make it more straightforward following the recommendations. The related work section is clearer to follow and more related to the paper's main contribution. They also included a handful of more recent related works.

Experimental design

The authors conducted additional experiments to support the claims following the recommendations.

Validity of the findings

By comparing with LLM baselines, the authors can demonstrate the effectiveness of their approach.

Additional comments

Ln 266. "To detect hallucinations in the generated rationales, we compare them against human-annotated rationales from a reference dataset (HateXplain) using similarity metrics."

To my knowledge, hallucination refers to the model generating information that's not entailed in the input, not the human-annotated span. The model may look at a distinct piece of evidence. As long as it's consistent with the input, it's fine.

More important problem. When predicting unseen text, there's no human-labeled rationale. The proposed iterative refinement won't work.

---

## Round 0.3 · accepted · Accept

The paper can now be accepted for publication in terms of validity and correctness of its content.

However, two minor formatting need resolving, which can be addressed during the publication process: (i) line 183 contains several missing references as '?' which need fixing, and (ii) the paragraph after line 367 is wrongly formatted and is not readable (after the URL), which also needs fixing.